# Infant diarrheal disease in rhesus macaques impedes microbiome maturation and is linked to uncultured *Campylobacter* species

Nicholas S. Rhoades[1,2], Isaac R. Cinco[2], Sara M. Hendrickson[3], Kamm Prongay[4], Andrew J. Haertel [4], Gilberto E. Flores [5], Mark K. Slifka [3] & Ilhem Messaoudi [2✉]

Diarrheal diseases remain one of the leading causes of death for children under 5 globally, disproportionately impacting those living in low- and middle-income countries (LMIC). *Campylobacter spp.*, a zoonotic pathogen, is one of the leading causes of food-borne infection in humans. Yet to be cultured *Campylobacter spp.* contribute to the total burden in diarrheal disease in children living in LMIC thus hampering interventions. We performed microbiome profiling and metagenomic genome assembly on samples collected from over 100 infant rhesus macaques longitudinally and during cases of clinical diarrhea within the first year of life. Acute diarrhea was associated with long-lasting taxonomic and functional shifts of the infant gut microbiome indicative of microbiome immaturity. We constructed 36 *Campylobacter* metagenomic assembled genomes (MAGs), many of which fell within 4 yet to be cultured species. Finally, we compared the uncultured *Campylobacter* MAGs assembled from infant macaques with publicly available human metagenomes to show that these uncultured species are also found in human fecal samples from LMIC. These data highlight the importance of unculturable *Campylobacter spp.* as an important target for reducing disease burden in LMIC children.

[1] Department of Molecular biology and Biochemistry, University of California Irvine, Irvine, CA, USA. [2] Department of Microbiology, Immunology and Molecular Genetics, College of Medicine, University of Kentucky, Lexington, KY, USA. [3] Division of Neuroscience, Oregon National Primate Research Center, Portland, OR, USA. [4] Division of Animal Resources and Research Support, Oregon National Primate Research Center, Oregon Health and Science University West Campus, Portland, OR, USA. [5] Department of Biology, California State University, Northridge, Northridge, CA, USA. ✉email: ilhem.messaoudi@uky.edu

Diarrheal diseases are the second leading cause of death in children under five years of age with over 500,000 fatalities every year that disproportionally impact low- and middle-income countries (LMIC)[1,2]. A wide range of enteric pathogens (*Rotavirus*, *Norovirus*, *Campylobacter*, *Shigella*, *Giardia*, etc.) are responsible for diarrheal diseases in children. However, these known pathogens only account for 40–50% of cases in LMIC leaving a substantial number of cases that cannot be attributed to a specific pathogen[3,4]. It also remains unclear why children who grow up in the same environment have differing susceptibility to enteric pathogens. One potential mechanism is the diversity of the gut microbiome which is highly dynamic in early life. It is well-established that the gut microbiome modulates host susceptibility to enteric pathogens through production of anti-microbial products, competition for niche space, and education of the immune system[5–8]. However, the role of specific components of the commensal gut microbiome in regulating susceptibility to and severity of diarrheal disease remains poorly understood due to lack of a clear definition of a "healthy" microbiome, a lack of longitudinal studies that include presymptom samples, and difficulty in collecting samples in resource-limited settings where diarrheal diseases are prevalent. A better understanding of the role of the gut microbial community would pave the way to the development of better therapeutic interventions and vaccination.

We have recently described an infant rhesus macaque model of environmentally acquired diarrheal disease and enteric dysfunction that is primarily driven by Campylobacterosis[9–11]. Our macaque model accurately recapitulates many of the key features of diarrheal diseases in resource limited settings including stunted developmental trajectory of the gut microbiome[9], markers of local and systemic immune dysfunction[10], growth faltering, and histopathological changes in the small and large intestine[11]. In this study, we utilize microbiome profiling and metagenomic genome assembly on longitudinal samples collected from over 100 infant macaques across 5 pre-determined time-points during their first year of life. Moreover, over the course of the study, 34 animals developed diarrhea, and in those instances, additional samples were collected at the time of entry and exit from the clinic and subjected to the same analyses. As observed in humans[12–14], diarrhea was associated with delayed maturation of the infant gut microbiome, specifically by the reduction of diversity in the gut microbial community which persisted after resolution of symptoms. Interestingly, infants that later developed diarrhea harbored a distinct microbiome from that of infants that remained asymptomatic as early as at 1 month of age. One of the distinguishing features was higher relative abundance of *Campylobacter* at the genus level that was evident before disease onset.

*Campylobacter* is one of the four leading causes of diarrheal diseases, and the primary cause of foodborne infections in humans globally, with the majority of infections being caused by the species *Campylobacter jejuni* (*C. jejuni*) and *Campylobacter coli* (*C. coli*)[15]. Campylobacterosis can lead to significant morbidity and mortality in children under five[1,2,16] especially those living in LMIC[17–20]. More importantly, children in LMIC are more likely to experience subclinical *Campylobacter* colonization, which has been associated with growth stunting, aberrant inflammation, and dysbiosis[21,22]. Despite its global impact on public health, our understanding of the complex etiology and true diversity within the *Campylobacter* genus remains incomplete. Over the past decade more than a dozen named species have been added to the genus and associated with multiple clinical manifestations including ulcerative colitis, periodontitis, diarrhea, and cancer[23–25]. Moreover, recent studies using culture-independent techniques such as qPCR and metagenomic genome assembly have identified additional non-*coli*/*jejuni Campylobacter* species

that had been missed by traditional culture assays, notably *Candidatus Campylobacter infans* (*C. infans*) which was initially assembled from a child experiencing diarrhea in India and has since been identified in samples from 6 additional LMIC countries[26–28]. In this study, we constructed 36 *Campylobacter* metagenomic assembled genomes (MAGs), 15 of which belong to uncultured species that could be classified into 4 uncultured metagenomic groups (UMGs). These uncultured *Campylobacter* species had unique metabolic and pathogenicity profiles, and the majority were previously reported in human samples. These data will aid in the efforts aimed at the isolation and cultivation of undescribed *Campylobacter* and provides genomic insights into their impact on human health.

## Results

**Diarrhea disrupts the taxonomic development of the infant macaque gut microbiome.** 16S rRNA gene amplicon sequencing was used to profile the fecal microbiome of 130 infant rhesus macaques, 34 of which developed clinical diarrhea during the first year of life (26.4%). Rectal swabs were collected at 1, 3, 6, 9 and 12 months of age (Fig. 1A). Additional swabs were collected any time an infant was admitted to the clinic for diarrhea and when the infant was released after treatment (Fig. 1A). The majority of infants in this study experienced their first case of clinical diarrhea between 3 and 9 months of age while only two of 34 infants experienced clinical diarrhea prior to 3-months (Fig. 1B, Table S1). We first examined the maturation of gut microbial communities of healthy infants that did not experience an episode of diarrhea. While *Prevotella* was highly abundant at all the timepoints, loss of milk degrading bacteria such as *Bifidobacterium* and concomitant increase in fiber degraders like *Treponema* was observed over the 12 months of life (Fig. 1C, D, Table S2). Additionally, the microbiome of infants that remained healthy progressively became closer to that of adult dams and the intergroup variability was decreased over the 12-month period (Fig. 1D–F).

However, disruptions in this developmental trajectory were observed in infants that developed diarrhea (none of the samples included in this analysis were obtained during acute disease). Specifically, a stunting in the trajectory was evident at 3-months of age, prior to most infants experiencing their first episode of diarrhea (Fig. 1D). These changes included increased dissimilarity from the dam gut microbiome (Fig. 1E) and increased within group dissimilarity (Fig. 1F) compared to healthy infants. This delay in maturation became more pronounced with increasing age (Fig. 1D, E). Moreover, the microbiome of sick infants was less phylogenetically diverse at the 6- to 12-month timepoints compared to that of infants that remained asymptomatic (Fig. 1G). Diversity of the microbial community was further reduced at the time of acute diarrhea as expected with an enteric infection (Fig. 1G). Although the diversity of the microbial community was significantly increased after treatment and exit from the clinic, it remained below that observed in healthy infants (Fig. 1G).

Taxonomically, we found no significant difference at the 16s amplicon level at the 1-month time point between animals that remained healthy and those that developed diarrhea (Fig. 1H). However, several significant differences were identified at the 3-month time point (when only 2 of 34 animals experienced their first episode) and beyond (Fig. 1H, Table S3). These changes included a sustained increase in known short chain fatty acid producers and fiber degraders in healthy infants, notably *Ruminococcaceae*, *Fibrobacter* and *Treponema* in infants that remained healthy (Fig. 1H, Table S3). On the other hand, the gut microbiomes of infants that developed diarrhea were enriched in

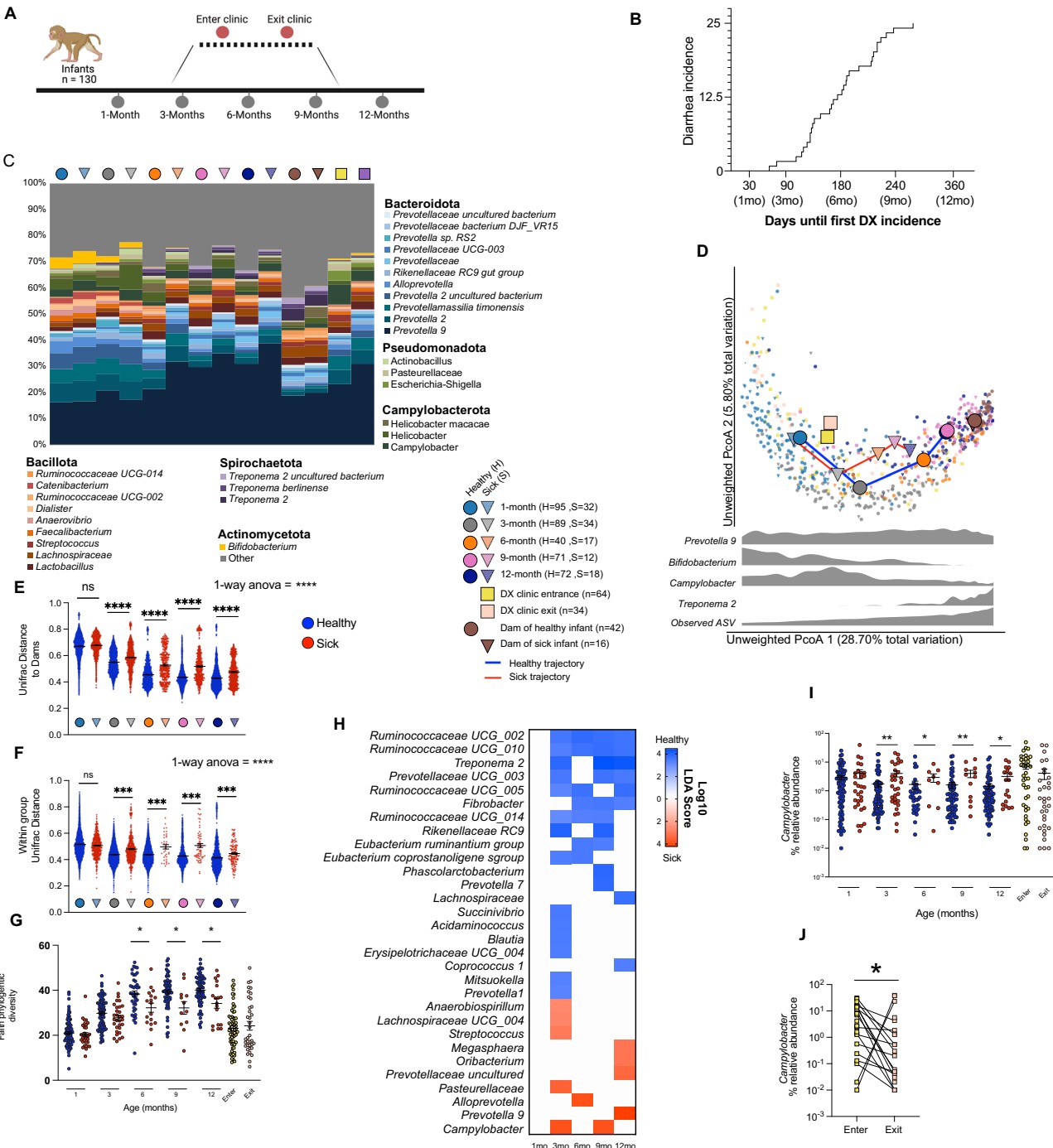

**Fig. 1 Diarrhea stunts the development of the infant microbiome. A** Study timeline and sample collection. The dotted line between 3 and 9 months denotes the period in which most of the infants entered and exited the clinic for treatment and sample collection during their first case of diarrhea. **B** Kaplan Meier curve of diarrhea incidence within the study population indicating the first incidence of clinical diarrhea for every animal that experienced symptoms over the course of the study (first 12 month of life). **C** Stacked bar plot of the top 30 most abundant taxa across all samples, each bar represents the average relative abundance of each taxon within the indicated group. **D** Principal coordinate analysis (PCoA) of fecal microbiome unweighted UniFrac distance (16S amplicon) colored by host status and timepoint. Small points represent individual samples while larger points represent the centroid for a given timepoint/host status. Solid lines connecting centroids illustrate the developmental trajectory of both healthy and sick infants. Density plots of key taxa and observed ASVs are in grey below the plot. **E–G** Dot plot of (**E**) UniFrac distance between dams and each infant timepoint/health status, **F** within group UniFrac distance for each infant group, and (**G**) Faith's phylogenetic diversity, in healthy and sick infants. **H** Heatmap of select differentially abundant taxa at longitudinal time-points as determined by LEfSe ($Log_{10}$ LDA score > 2). **I** Longitudinal dot plot of *Campylobacter* relative abundance, in healthy and sick infants. **J** Relative abundance of *Campylobacter* at first clinic entry and exit. Significance for panels (**E**) and (**F**) was determined using one-way ANOVA ****$p < 0.0001$, with Holm-Sidak's multiple comparison test, ****$p < 0.0001$, error bars = SEM. Significance for panels (**G**) and (**I**) was determined using unpaired *T*-test at each time-point, *$p < 0.05$, **$p < 0.01$, ***$p < 0.001$, error bars = SEM. Significance for panels (**J**) was determined using paired *T*-test, *$p < 0.05$. DX Diarrhea.

*Prevotella, Streptococcus*, and *Campylobacter* species (Fig. 1H, Table S3). Indeed, a significantly higher relative abundance of *Campylobacter* was observed at timepoints 3- to 12-months in infants that developed diarrhea (Fig. 1I). The relative abundance of *Campylobacter* was significantly higher when sick infants entered the clinic with acute diarrhea compared to their exit (Fig. 1J). Although, the relative abundance of *Campylobacter* decreased at discharge, it remained higher than that observed in healthy infants (Fig. 1J). Of note, Campylobacter relative abundance assessed by 16S rRNA amplicon sequencing correlated significantly with *Campylobacter* colony forming units as measured by qPCR (Fig. S1A). Together these data suggest that diarrhea stunts the development of the infant gut microbiome resulting in a more heterogeneous but less diverse community with increased *Campylobacter* burden.

We also profiled the gut mycobiome and its maturation over the first year of life and how it is impacted by diarrheal disease using internal transcribed sequence (ITS) sequencing using a subset of samples (Fig. S1B). As established for the prokaryotic microbiome, the gut mycobiome became more diverse over time (Fig. S1B–D). While we observed no differences between the two groups at the 1-month timepoint, animals with a history of diarrhea exhibited a less diverse mycobiome than age-matched healthy infants during acute disease and at 12-months of age (Fig. S1D). Taxonomically, the macaque mycobiome is poorly defined, leading to a large portion of sequences being classified as "unidentified fungi" (Fig. S1B). Nevertheless, differentially abundant taxa (*Rhizophydium* and *Metschnikowia continentalis*) were identified during acute disease and at 12-months of age (Fig. S1E).

A majority of the infants in this study were vaccinated against *C. coli* as previously described[29]. While *C. coli* vaccination reduced overall diarrheal disease burden, it did not impact the composition of the fecal microbiome[29]. However, since *Campylobacter spp.* were associated with diarrheal disease in our study population, we conducted additional analyses to confirm that the trends observed were not associated with vaccination. Microbiome stunting, reduced community richness, and increased *Campylobacter* abundance were still observed when samples were subsetted based on vaccination status (Fig. S2A–J).

**Role of maternal microbiome in infant susceptibility to diarrheal diseases**. The role of the maternal microbiome in modulating infant susceptibility to diarrheal diseases remains poorly defined. Therefore, we compared the microbiomes of dams that gave birth to infants that later developed diarrhea to those that gave birth to infants that remained asymptomatic at ~1 month after delivery. At the 16S rRNA amplicon level, minor differences were noted in the overall composition of the gut microbiome between these two groups (Fig. S3A, B). However, higher resolution taxonomic data generated using shotgun metagenomics indicated that the microbiome of dams who gave birth to infants that later experienced diarrhea harbored increased abundance of both *Escherichia coli* (*E. coli*), *Bifidobacterium pseudocatenulatum* (*B. pseudocatenulatum*), and *C. jejuni* (Fig. S3C, D). Additionally, the gut microbiome of dams whose infants later developed diarrhea was enriched in nitrate reduction and hexitol fermentation pathways while that of dams whose infants remained healthy was enriched in homolactic fermentation (Fig. S3E, F).

**Diarrheal disease disrupts the functional maturation of the infant macaque gut microbiome**. We next used shotgun metagenomics to define the development of the infant gut microbiome at the functional and taxonomic levels and determine how diarrhea disrupts these processes. As seen with 16S amplicon data,

age was a strong predictor of gut microbiome functional potential and taxonomic composition (Fig. S4A, B). Specifically, at early time-points the microbiome was enriched in pathways for the degradation of sugars commonly found in breastmilk including fucose degradation, Bifidobacterium shunt (1-month) and 1,5-anhydro-fructose degradation (3-months) (Fig. S4C). At later time points anabolic pathways such as L-isoleucine biosynthesis (6-months), L-tryptophan biosynthesis and folate transformation (9-months) as well as pathways associated with the production of short-chain fatty acids such as pyruvate fermentation to iso-butanol (9-months) were increased (Fig. S4C).

Next, we explored potential predictors of diarrheal disease and assessed the impact of diarrhea on the functional potential of the gut microbiome (Fig. 2A). Although overall functional capacity did not differ between infants that remained asymptomatic and those that developed diarrhea at 1 month of age (prior to any diarrhea incident; pre-DX) (Fig. 2A), several metabolic pathways were differentially abundant between these two groups at this early time point. Specifically, glycolysis, and homolactic fermentation pathways were more abundant in infants that remained asymptomatic while anhydro-fructose and amino-butanate degradation pathways were enriched in infants that later developed diarrhea (Fig. 2B, Table S4). To assess the impact of acute diarrhea and ensuing treatment, we grouped animals into early (3–6 months; Early-DX) and late onset (6.1–11 months; Late-DX) of disease. At both time frames, the microbiomes of healthy infants were enriched in L-arginine biosynthesis and folate transformation (Fig. 2B, Table S4). Acute diarrhea at 3-6 months of age led to increased abundance of fatty acid elongation, palmitoleate biosynthesis, and phosphatidylcholine editing pathways (Fig. 2B, Table S4). After clinic exit, the microbiomes of these infants were enriched in nitrate reduction and histidine degradation pathways (Fig. 2B, Table S4). The microbiomes of animals that developed diarrhea between 6 and 11 months of age were enriched in fatty acid elongation during acute disease and sulfate reduction and sucrose degradation upon exiting the clinic (Fig. 2B, Table S4).

At 9 or 12 months of age (at least 1 month after the last diarrhea episode; Post-DX) the microbiomes of animals that experienced diarrhea were enriched in multiple pathways including thiazole biosynthesis, methyl-citrate cycle, palmitoleate biosynthesis, and fucose degradation (Fig. 2B). Since the pathway for folate transformation was enriched in healthy infants across multiple time-points, we attempted to identify the species contributing to this pathway. This analysis revealed that *Bracyspira pilosicoli* was a major contributor to folate transformation in healthy infants early in life and replaced by multiple *Prevotella* species namely AM42_24 and CAG_520 later in life (Fig. 2C).

**Disruption in the taxonomic maturation of the gut microbiome due to diarrheal disease**. Taxonomically, we observed a loss of all *Bifidobacterium* species and concomitant increase of *Treponema* at ~6-months of age which coincided with when most infant macaques are weaned (Fig. S4B, D). Additionally, we found that Prevotella species were dynamic at the species level, with *Prevotella copri* (*P. copri*) only becoming dominant at ~6month of age, while uncultured *Prevotella sp_885* was only observed in young infants and *Prevotella* CAG_5226 was only abundant in the dams (Fig. S4D). At 1 month of age (pre), although overall taxonomic composition did not differ between animals that remained asymptomatic and those that developed diarrhea (Fig. 3A), multiple species were differentially abundant between the two groups (Fig. 3B). Of note, relative abundance of *Ruminococcus* CAG_624 was increased in infants that remained asymptomatic while relative abundance of *Anaerostipes hadrus*

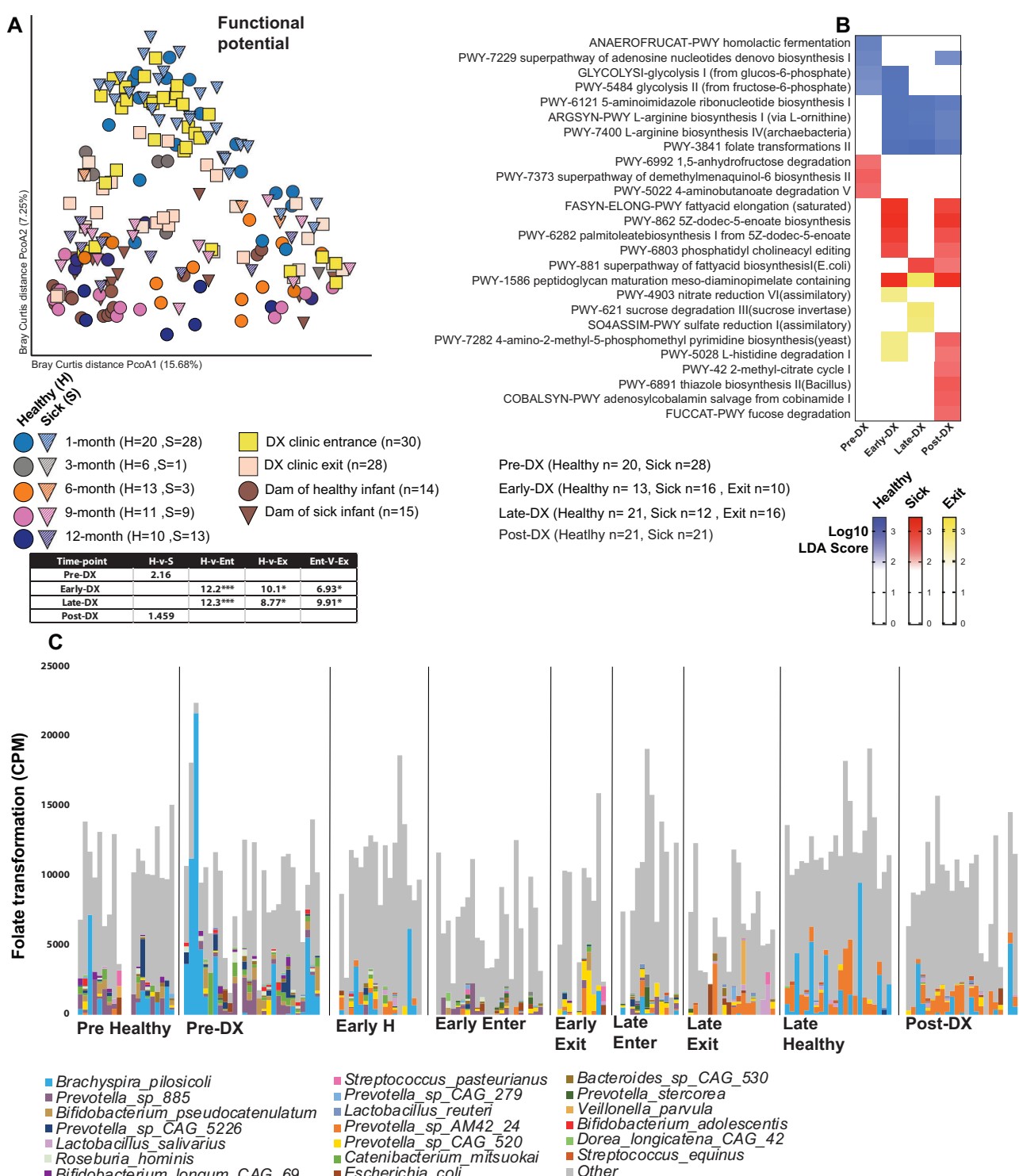

**Fig. 2 Impact of acute and distant diarrhea on the functional potential of the gut microbiome. A** PCoA of Bray-Curtis dissimilarity built on the abundance of all functional genes annotated using HUMAnN3 and the Uniref90 database colored by host status and timepoint. Significance and variance by host status for each time-point determined by PERMANOVA and reported in the included table. **B** Heatmap of select differentially abundant functional pathway at each time-point as determined by LEfSe (Log$_{10}$ LDA score > 2). **C** Bar-plot of folate transformation pathway abundance, stratified by bacterial species to which that pathway could be assigned. Each stacked bar represents an individual sample at the indicated time-points. DX Diarrhea.

was increased in infants that later developed diarrhea (Fig. 3B, Table S5).

We then assessed the impact of acute diarrhea and ensuing treatment in infants that developed diarrhea 3–6 months (early) and 6.1–11 months (late) of age. In both timeframes, the microbiomes of infants that remained asymptomatic were enriched in *Lactobacillus*, *Treponema*, *Bracyspira* species as well as *Helicobacter macacae* (Fig. 3B, Table S5). In contrast, acute diarrhea was associated with increased abundance of multiple *Campylobacter* and *Prevotella* species (Fig. 3B, Table S5). After exit from clinic, the relative abundance of *P. copri*, *Streptococcus lutetuensis*, and *Veillonella parvula* was increased (Fig. 3B,

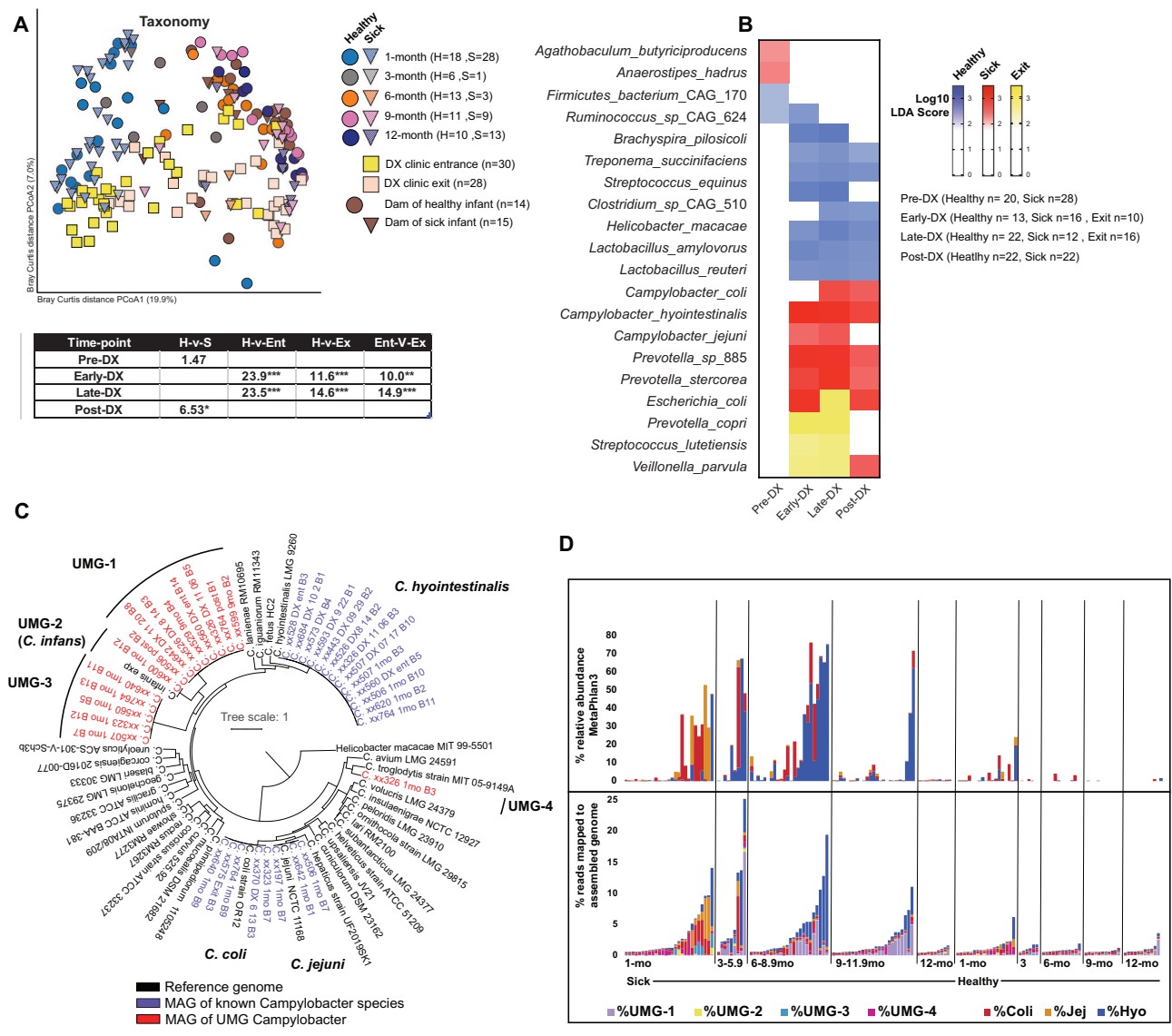

**Fig. 3 Impact of diarrhea on the species level composition of the gut microbiome. A** PCoA of Bray-Curtis dissimilarity built on the species level taxonomy generated using MetaPhlAn and colored by host status and timepoint. **B** Heatmap of select differentially abundant species pathway at each time-point as determined by LEfSe (Log$_{10}$ LDA score > 2). **C** Core gene phylogenetic tree built using 60 randomly selected conserved cross-genus gene families (PGfams) for metagenomically assembled *Campylobacter* genomes (MAGs) and 32 known *Campylobacter* species. Black = Reference genome, Blue = MAG similar to known species, Red = MAG not similar to any known species. **D** Stacked bars of *Campylobacter* relative abundance as measured by Metaphlan3 (top) and mapping to MAGs and reference genomes (bottom). Each vertical bar represents a single sample at the denoted timepoint/health status. For "Sick" animals samples collected upon exit from the clinic were not included. DX Diarrhea.

Table S5). At 12-months of age (at least 1-month after the last exit from clinic), many differences that were observed during acute disease persisted, most notably the increased abundance of *Campylobacter* persisted (Fig. 3B, Table S5).

**Metagenomic genome assembly reveals a diverse set of *Campylobacter* species associated with diarrhea.** Given the increased relative abundance of *Campylobacter* in infants who experienced diarrhea and the recent reports of the critical role that unrecognized *Campylobacter* species play in childhood diarrheal disease and growth/cognitive stunting[27,28,30], we utilized metagenomic genome assembly to explore the diversity of enteric *Campylobacter* species within infant macaques. We assembled 36 high quality (>80% complete with <2% contamination) *Campylobacter* metagenomic assembled genomes (MAGs). Despite the inclusion of 58 samples from healthy infants, all *Campylobacter* MAGs

originated from samples collected from infants that developed diarrhea. A phylogenetic tree based on the alignment of 50 conserved single copy genes showed that most MAG belonged to known species including 13 *Campylobacter hyointestinalis* (*C. hyointestinalis*) genomes, 5 *C. coli* genomes, and 3 *C. jejuni* genomes (Fig. 3C, Tables S6, S7). The remaining 15 *Campylobacter* MAGs were divided into 4 distinct groups (UMGs) with 8 genomes falling into "UMG-1", 1 genome in "UMG-2", 5 genomes in "UMG-3", and 1 in "UMG-4" (Fig. 3C, Table S6). UMG 2 was closely related to the recently identified MAG *C. infans* that was assembled from a child with acute severe diarrhea in the global enterics multi-center study (GEMS)[27]. UMGs 1 and 3 were closely related to each other and to *C. hyointestinalis* (Fig. 3C, Table S6). UMG-4 was distinct and most closely related to *Campylobacter troglodytis* (*C. troglodytis*) which was isolated from chimpanzees[31].

Next, we determined if these uncultered species could be detected using a traditional shotgun metagenomics taxonomic

classifier MetaPhlan3[32]. This method was able to assign a relative abundance to the 3 known *Campylobacter* species (*C. hyointestinalis, C. coli,* and *C. jejuni*) but was unable to quantify the 4 *Campylobacter* UMG (Fig. 3D). We then quantified reads that map to the assembled *Campylobacter* MAGs including the 4 UMGs using BowTie2 (Fig. 3D). This analysis reveals that the 4 *Campylobacter* UMGs were prevalent in sick infant macaques (Fig. 3D). These analyses revealed that infant macaques are colonized by a diverse set of both known and unknown *Campylobacter* species that are more abundant in infants that will or have already experienced an episode of diarrhea.

**Genomic diversity of "Unknown" *Campylobacter* species**. To explore the genomic diversity of the *Campylobacter* MAGs identified in this study relative to published *Campylobacter* MAG, we performed a pangenomic analysis using Anvi'o[33]. We included genomes from 18 reference *Campylobacter* species, 11 recently described MAGs and genomes from rhesus macaques[9,10,34], and 11 human MAGs[35,36] including *C. infans*[27]. All human MAGs fell within UMG-2, -3 and -4 based on average nucleotide identity and several were closely related to MAGs assembled from infants that developed diarrhea in this study, highlighting the relevance of the rhesus macaque model for investigating these uncultured *Campylobacter* species (Fig. 4A–C, Table S6).

Moreover, these humans MAGs were identified from samples obtained from LMIC where infant diarrhea is more prevalent[35,36] (Fig. 4D). We identified a total of 18,076 gene clusters (GCs), with 7099 GCs being found in only a single genome (Singletons), 3078 GCs being found in 2–3 genomes (Accessory), and 69 GCs that were shared across all 77 genomes (Fig. 4B). We also identified 408 GCs that were present in at least 90% of all genomes. This expanded core included GCs essential for amino acid transport (*hisM, hisC, abcC*) and metabolism (*pykF, pgk, gltA*) (Fig. 4B).

**Motility and adhesion factors in yet to be cultured *Campylobacter* MAG**. Next, we investigated the presence of known virulence factors in the *Campylobacter* UMG identified in this study. Using Anvi'o we assigned Clusters of Orthologous Groups (COG) annotations to GC's and conducted a functional enrichment analysis. *Campylobacter* are highly motile due to a single polar flagellum at both poles of the cell body which allows them to burrow through host mucosa and access the epithelial surface where they attach and invade. Therefore, we first examined GC's that played a role in motility, chemotaxis, and adhesion to host cells. Most genes associated with flagellar assembly (all *flg, fli, mot* genes) were highly conserved across all genomes except for UMG-3 (Fig. S5A). Additionally, MAGs within UMG-3 lacked genes associated with chemotaxis that were highly conserved across all other species (Fig. S5A). Next, we examined additional key host cell adhesion and invasion genes (*pldA, ciaB, cadF*). While *pldA*, which is important for colonization of the avian cecum[37], was only found in UMG-3 and -4, *cadF* and *ciaB*, which bind host epithelial fibronectin and facilitate internalization, were found in UMG-1 and -2. Finally, UMG-3 only harbored *ciaB* (Fig. S5A).

At the amino acid level, the *cadF* and *ciaB* sequences from rhesus MAGs of *C. jejuni, C. coli,* and *C. hyointestinalis* were nearly identical to their respective human reference genomes (Fig. S5B, C). On the other hand, the *cadF* sequences of UMG-1 and -2 were phylogenetically distinct from any other known *Campylobacter* species, with one rhesus macaque MAG *cadF* sequence nearly identical to that of *C. infans* and another human MAG (Fig. S5B). A Similar pattern was observed for *ciaB* in UMG-2 (Fig. S5C). However, the ciaB amino acid sequences of

UMG-3 from humans and rhesus macaques were distinct (Fig. S5C). These data show that the *Campylobacter* UMG, with the exception of UMG-3, have the genetic capacity for motility and chemotaxis as well as a varied ability to attach and invade the host epithelia.

**Additional virulence factors in yet to be cultured *Campylobacter***. Other traits of *Campylobacter* pathogenesis include iron acquisition[38], toxin production, and antimicrobial resistance[39]. Using data from our functional enrichment analysis and annotations generated using the Pathosystems Resource Integration Center (PATRIC), we further probed the UMGs for virulence factors associated with these processes. Enterochelin transport system (*ceuA-D*), is an iron acquisition system that is highly conserved in *C. coli* and *jejuni*. This transport system was absent in all 4 *Campylobacter* UMG (Fig. 5A). However, the UMG obtained from both humans and rhesus macaques possessed other genes for iron acquisition including *FeoA, HemH,* and *HugZ* (Fig. 5A).

Additionally, we examined the *Liv* operon, a key set of genes involved in the import of the branch chain amino acids leucine, isoleucine, and valine, that is essential for the colonization of the avian caecum[40]. This operon was present in all *C. coli* and *C. jejuni* genomes and all *C. hyointestinalis* except for the *LivJ* gene (Fig. 5A). Interestingly, these genes were absent in all *Campylobacter* MAGs except for those within UMG-2 (*C.infans*) which, like *C. hyointestinalis*, only lacked the *LivJ* gene (Fig. 5A). Finally, we examined a subset of oxidative stress genes important for survival of *Campylobacter* in microaerophilic conditions including catalase and superoxide dismutase[41]. All known and UMG *Campylobacter* species had at least one of these oxidative stress genes suggesting some capacity to neutralize reactive oxygen species (Fig. 5A).

While *Campylobacter* produces a multitude of toxins, only cytolethal distending toxin (CDT) has been studied in detail[15]. All 3 subunits, encoded by *cdtA-C* are required for toxicity and host cell death[42]. Interestingly, the representative genomes of both *C. coli* and *C. hyointestinalis* selected for this analysis lack *cdtA* and *cdtC* while all rhesus macaque isolates, and MAGs encode all 3 genes (Fig. 5B, Table S8). All 3 subunits were present in both the reference genome and rhesus macaque MAGs of *C. jejuni* (Fig. 5B). Finally, while the complete *cdt* gene set was present in 9 of 27 other known non-Coli/Jejuni *Campylobacter* species, these genes were absent in all UMG (Fig. 5B).

The *Campylobacter* Multidrug Efflux system (CME) plays an essential role in the resistance of *C. coli* and *jejuni* to a multitude of antimicrobial agents[43]. CME efflux pumps are composed of two distinct systems, *cmeABC* encode the primary efflux pump while *cmeDEF* encodes a secondary efflux pump that provides additional resistance but not to the level of *cmeABC*[44]. The transcription of both efflux pumps is under regulation of *cmeR*. Both *C. coli* and *C. jejuni* genomes encode all 7 genes (Fig. 5C). Although *C. hyointestinalis* MAGs from rhesus macaques encode the *cmeABC* system, the human reference genome of *C. hyointestinalis* is void of all cme genes (Fig. 5C). With the exception of MAGs within UMG-4, which encoded the *cmeDEF* genes, CMW genes were absent in all UMG (Fig. 5C).

Finally, we examined key genes involved in respiration and more specifically the electron transport chain. *Campylobacter* utilize unique and essential electron donors and acceptors such as Formate and Fumarate. Key mutations in the *nuoC-E* genes of respiratory Complex I (NADH ubiquinone oxidoreductase) found in *C. coli, C. jejuni,* and many other cultured *Campylobacter* facilitate the use of flavodoxin and menaquinone rather than NADH and ubiquinone as electron carriers to generate

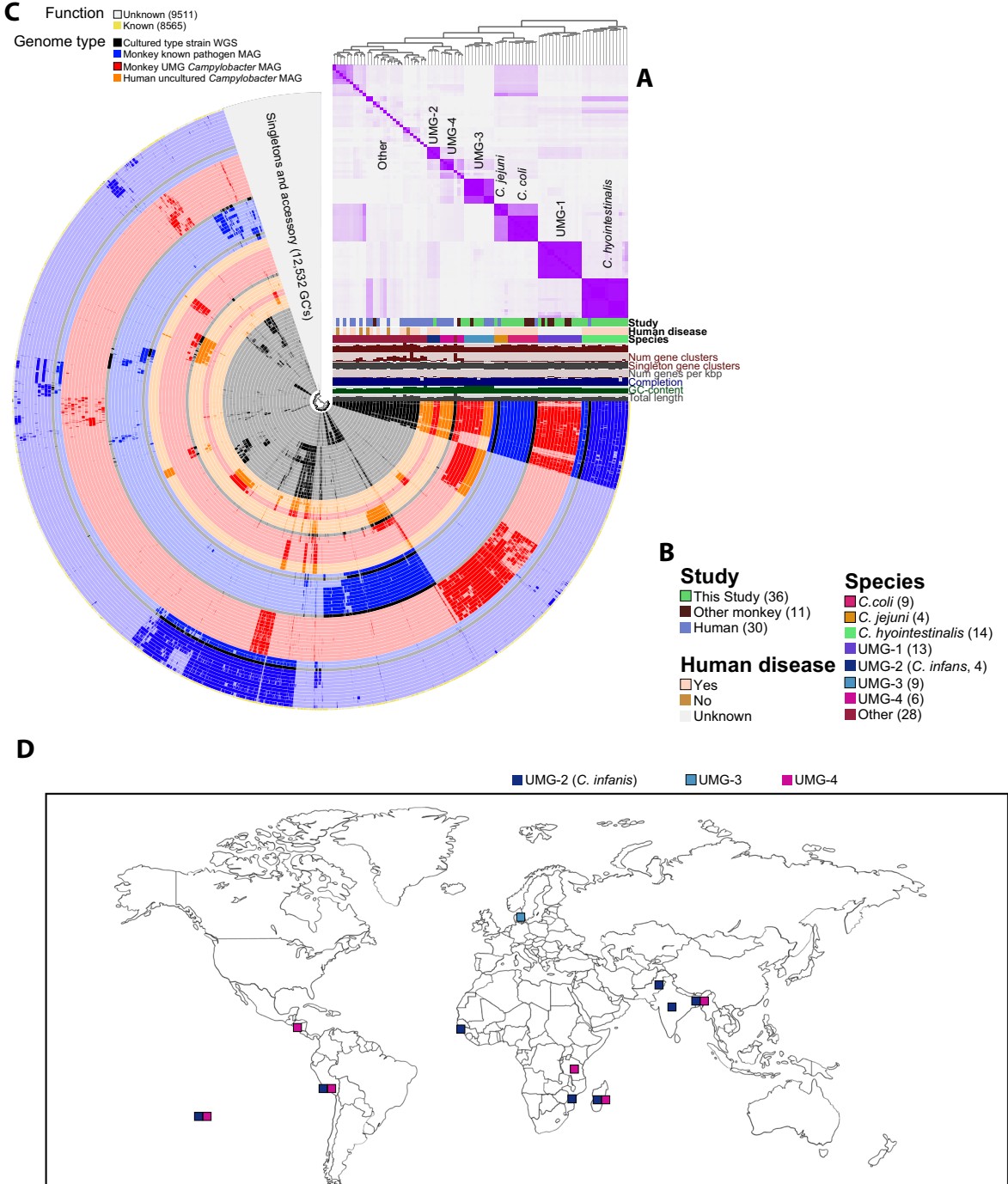

**Fig. 4 A diverse set of Campylobacter are associated with diarrheal disease in infant macaques revealed by genome assembly. A** Heatmap of average nucleotide identity (ANI) between all *Campylobacter* genomes in the pangenomic analysis. **B** Metadata and information about genomes included in analysis. **C** Pangenome of 77 *Campylobacter* genomes generated using Anvi'o. Each concentric circle represents a bacterial genome: reference genome (Black), MAGs assembled from rhesus macaques that are similar to a known species (Blue), MAGs assembled from rhesus macaques that are not similar to a known species (UMG) (Red), MAGs assembled from human metagenomes that are similar to our *Campylobacter* UMG genomes (Orange). Blank areas in each circle indicate the absence of a particular gene cluster (GC) in that genome. Outer ring denotes GCs with an annotated function COG function (Yellow). Genomes are ordered by ANI, as depicted by the red heatmap in panel (**A**). **D** Global map illustrating the distribution of UMGs 2–4 in human data for UMG-2 derived from PCR screening by Bian et al.[27] Additional distribution information for UMGs 2–4 based on locations of samples collected from which human UMG MAGs were assembled in by Almedia et. al and Nayfach et al.[35,36]

membrane potential[45,46]. This pattern can be used to distinguish between *C. coli/jejuni* and *C. hyointestinalis/fetus* which use the canonical NADH and ubiquinone as their electron transporters. MAGs from rhesus macaques and genomes from *C. coli/jejuni* and *C. hyointestinalis* isolates recapitulated this pattern (Fig. 5D).

In contrast, all MAGs belonging to UMGs 1–4 lacked all genes needed to assemble respiratory Complex I (NADH ubiquinone oxidoreductase) (Fig. 5D). This was also observed in reference genomes from *C. showae*, *C. mucosalis*, *C. rectus*, *C. avium*, and *C. troglodytis* (Fig. 5D).

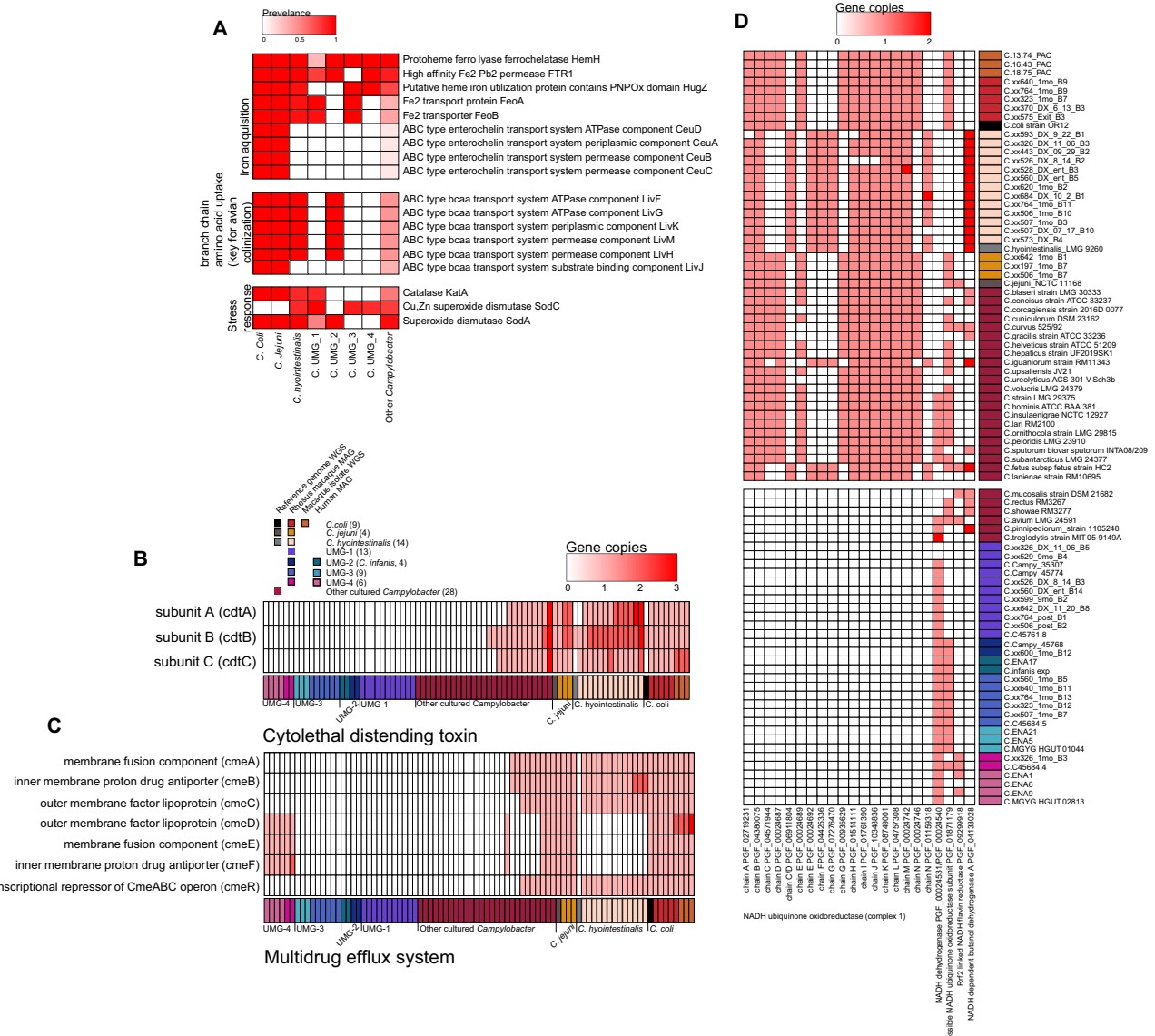

**Fig. 5 Iron acquisition, toxin production, antimicrobial resistance and energy generation in Campylobacter UMGs. A** Prevalence heatmap of genes necessary for iron acquisition, branch-chain amino acid uptake, and response to oxidative stress, with the color of each box showing the prevalence of that gene within the noted genome group. **B–D** Prevalence heatmaps for cytolethal distending toxin (**B**), *Campylobacter* multidrug efflux genes (**C**), and electron transport chain complex I (**D**) split by individual genome.

## Discussion

It is widely accepted that the gut microbiome plays a key role in modulating host susceptibility to enteric disease[3,4,12–14]. However, microbiome studies are challenging to conduct in humans in part due to the substantive individual variability of the human gut microbiome and the difficulties of longitudinal sampling before and after disease. In this study, we leveraged an infant rhesus macaque model to define the maturation/evolution of the gut microbiome throughout the first year of life as well as during and at the end of acute diarrhea episodes.

Data presented in this manuscript recapitulate the pattern of microbiome stunting that has been observed in humans with diarrheal disease[14,47]. This disruption was associated with increased intra-group variability and a reduced diversity of the gut community. The divergence between microbial communities of animals that remained asymptomatic and those that developed diarrhea was evident as early as 1–3 month of age before any clinical episode. Moreover, relative abundance of *Campylobacter* was increased as early as 3-months of age in the microbiome of

infants that later developed diarrhea. These findings suggest that subclinical colonization with pathobionts may disrupt the maturation of the microbiome and precede clinical disease[3,13,48].

In addition to the prokaryotic microbiome, we also profiled the gut mycobiome of infant macaques. We report that the diversity of the mycobiome increased with age, a process that was disrupted by diarrhea, as evidenced by the abundance in rhizophydium and *metschnikowia continentalis* at 12-months of age. In humans, dysbiosis of the gut mycobiome has been reported in individuals with irritable bowel syndrome (IBS) and following antibiotic treatments[49,50]. Further studies are needed to determine if the mycobiome is truly disrupted by diarrheal disease or simply a co-variate of microbial dysbiosis.

We also found that the gut microbiome of dams that gave birth to infants who experienced diarrhea was both functionally and taxonomically distinct prior to the infants becoming sick, including increased pathogen burden (*C. jejuni* and *E. coli*). Our data align with recent findings from human cohorts that demonstrated that the maternal microbiome can modulate the

infant microbiome composition, immune development, and infant growth[51–53]. However, the mechanism by which the maternal microbiome influences protection and/or vulnerability to disease in the infant deserves further attention as the maternal microbiome is dynamic during the perinatal period.

Similar to what has been shown in humans, the microbiome of infant macaques that were still nursing was enriched in pathways associated with milk-oligosaccharide degradation such as fucose degradation[54]. Additionally, microbial pathways for the production of short-chain fatty acids were increased in post-weaned infants in agreement with observational data from humans[54,55]. As noted with taxonomic stunting, diarrhea also disrupted the functional maturation of the infant gut microbiome leading to a reduction in the abundance of folate transformation pathways and an increase in palmitoleate biosynthesis pathway among others. Folate, or vitamin B9, is among the essential vitamins that the host relies on the commensal microbes to synthesize and transform prior to absorption[56]. In line with this observation, reads mapping to some organisms known to be critical in this pathway such as several *Bifidobacterium* species (*B. pseudocatenulatum, B. adolescentis, B. longum*) were detected[57]. Together, these observations indicate that diarrheal disease disrupts the colonic microbiome's ability to transform folate, in line with histopathological finding from infant macaques with EED[11].

In agreement with our previous findings, we show that diarrhea was associated with increased abundance of the immunomodulatory palmitoleic acid production pathways[9,58]. Finally, we found that a history of diarrhea resulted in a gut microbiome enriched in pathways for the utilization of alternative electron acceptors such and Nitrate and Sulfate. Host inflammation has been shown to increase Nitrogen electron acceptor and antibiotic use can disrupt redox states within the gut[59]. It is possible that these shifts in redox dynamics prolong microbial dysbiosis and exacerbate inflammatory outcomes.

Taxonomically, we report decreased abundance of beneficial microbes (i.e., *Prevotella, Treponema succinifaciens* and *H. macacae*) in infants that experienced diarrhea across the entire timeline. In humans, operational taxonomic units of *Prevotella* have been reported to be overrepresented healthy infants and adults in countries outside of the United States of America[60] in agreement with our earlier report that the infant macaque microbiome was more closely related to that of children in LMIC[9]. Furthermore, *H. macacae* has previously been shown to have a potentially antagonistic association with *Campylobacter* likely through physical competition for resources in the colonic mucosa[9,61]. *Lactobacillus*, a prominent genus in the infant gut, was also reduced following diarrhea[62]. *Lactobacillus* and *Treponema* may play a similar role to *H. macacae* in humans[63,64]. In contrast, *Campylobacter spp.* were among the major taxa that were more abundant in infants that later experienced diarrhea. Due to the self-limiting nature of *Campylobacter* infection, it is probable that, following acute diarrhea, the relative abundance of these bacteria will decrease overtime. However, sick individuals may become asymptomatic carriers resulting in future reoccurrence or spread of the disease. In addition, total *Campylobacter* burden is higher than that of *C. jejuni* and *C. coli* combined in LMIC countries[26–28], suggesting an important role for other *Campylobacter* species. We and others have identified multiple MAG *Campylobacter* species from both rhesus macaques and humans that have yet to cultured[9,10,27]. Indeed, several studies have suggested an important role for these unidentified *Campylobacter* species in growth and cognitive stunting in children[28,65,66]. Here we identified 4 genetically distinct *Campylobacter* clusters (UMG) including 15 MAGs assembled in this study and 7 previously assembled MAGs[9,10,34]. We combined the MAGs detected in the infant macaque microbiome with 10 MAGs assembled from human samples that could not be classified to determine their relatedness[27,35,36].

The MAGs that we identified as UMG-2 shared significant genetic homology and likely belong to the same group as the recently identified candidate species *C. infans* that was assembled from the metagenome of a child experiencing severe diarrhea in India[27]. UMGs 3 and 4 contained genomes assembled from both human and rhesus macaques but their relationship to other known *Campylobacter* species had not been previously investigated. Using reported data sets[27,35,36], we found that MAG groups UMG-2 and UMG-4 were largely distributed in LMIC countries, suggesting a broader impact of uncultured *Campylobacter*. We investigated the genomic diversity of these unidentified *Campylobacter* MAGs, focusing on known virulence factors and metabolic processes. One caveat to this analysis is the fact that MAGs are often of lower quality that whole genomes sequences (WGS) generated from isolates. Indeed, MAGs are rarely assembled into a single genomic contig and therefore more likely to miss genes that are difficult to assemble from a mixed sample. Therefore, we focused our analysis on large groups of genes that encoded shared functions, notably flagellar assembly, adhesion, components of an efflux pump, iron acquisition and cellular respiration.

UMG-3 lacked all genes for chemotaxis and flagellar assembly suggesting that they are non-motile. This gene group is critical for the ability of pathogenic *Campylobacter* to invade the host mucosa and contact the host epithelia as well as secreting non-flagellar proteins that impact virulence[67,68]. Interestingly *C. gracilis*, and *C. hominis* also lack a flagellum but still cause periodontal disease and colonize the guts of children in the LMIC countries respectively[69–72]. Additional analyses showed that the amino acid sequences of *cadF* and *ciaB* in rhesus MAG were nearly identical to those of human reference genomes suggesting a broad ability of *Campylobacter* to colonize multiple species. Iron acquisition in *C. jejuni/coli* is primarily facilitated by a siderophore enterochelin scavenger system encoded by the *ceu* genes[73]. This system was found in both the human reference genome as well as *C. jejuni/coli* MAGs assembled from rhesus macaques but was absent in the genomes of nearly all other *Campylobacter* species including the 4 UMGs. However, the *Campylobacter* UMG contained alternative genes for iron acquisition (*hugZ, ftr1, feoA/B*)[74–76]. These finding suggest that our unknown UMGs use alternative iron acquisition methods which may impact their pathogenicity. Additionally, UMG-2/*C. infans* encoded genes in the *liv* operon, a branch chain amino acid uptake system that is essential for avian colonization[40]. The presence of these genes may indicate that UMG-2/*C. infans* may have an avian reservoir which provide an explanation for its global distribution.

UMGs lacked genes to produce cytolethal distending toxin (CDT), which has been shown to cleave host DNA resulting in cell death[77]. All three essential components were present in the *C. coli* and *C. hyointestinalis* genomes assembled from rhesus macaque samples. CDT is a key component of *Campylobacter* pathogenicity and its absence in UMGs suggest that they may have additional yet to be described toxins or modes of causing disease. Another key virulence factor is the ability to resist toxic compounds such as bile salts, heavy metals, or antimicrobial agents. This is primarily accomplished in *C. jejuni/coli* via are the *cme* Multidrug efflux pumps *cmeABC* and to a lesser extent *cmeDEF*[44,78] that confer resistance to multiple structurally unrelated antimicrobial compounds including rifampin, cephalothin, cefoperazone[79,80]. The *cme* genes were absent in the UMGs identified in this study with the exception of UMG-4, which encoded *cmeDEF*. This finding may explain why these organisms cannot be isolated on standard *Campylobacter* selective

media. However, since the CME pumps are the primary mechanism by which *C. jejuni/coli* tolerate bile salts[81], it is likely that the *Campylobacter* UMG encode a different efflux pump that could be challenging to identify without first cultivating these UMG species.

*C. jejuni/coli* harbor key mutations in the *nuoC-E* genes of respiratory Complex I (NADH ubiquinone oxidoreductase) that facilitate the use of flavodoxin and menaquinone rather than NADH and ubiquinone as electron carriers to generate membrane potential[45,46]. Similar isoforms of *nuoC-E* were detected in 19 additional reference *Campylobacter* species. Isoforms of *nuoC-E* genes necessary for the use of NADH and ubiquinone were only detected in *C. hyointestinalis*, *C. fetus*, *C. iguaniorum*, and *C. lanienae*. Surprisingly, UMGs 1–4 lacked the genes for a Complex I all together. Complex I genes are also absent in *C. avium*, *C. showae*, and *C. troglodytis*. The metabolic implications of this finding are unclear highlighting the need for additional studies.

Despite these findings, this study comes with some key limitations. Firstly, while our use of an infant macaque model allows for the control of many variables which typically confound microbiome studies, this model is not guaranteed to reflect clinical findings in LMIC infants. While similar to gut microbiome of human the macaque microbiome has its own unique constituent such as *H. macacae* which are not found in humans. Secondly, this study is limited to molecular methods such as metagenomic genome assembly. While these finding provide strong evidence for pathogenic, metabolic and physiological characteristic of the described UMG *Campylobacter*, these bacteria remain uncultured and therefore lack phenotypic support for our findings.

In summary, this study highlights the need to better understand the role of uncultured, *Campylobacter* species in childhood diarrheal disease burden. While our primary data are derived from infant macaques, the recent sharp increase in publicly available MAGs allowed us to trace these MAGs to human samples obtained from LMIC. This comparative genomic analysis paves the way to address this critical gap in our knowledge.

## Materials and methods

**Animal cohorts.** All rhesus macaque studies were overseen and approved by the OHSU/ONPRC Institutional Animal Care and Use Committees (IACUC) per the National Institutes of Health guide for the care and use of laboratory animals. Animals were housed per the standards established by the US Federal Animal Welfare Act and *The Guide for the Care and Use of Laboratory Animals*. We have complied with all relevant ethical regulations for animal use. All animals were tested for simian viruses (Simian Immunodeficiency Virus, Simian Retrovirus 2, Macacine herpesvirus 1, and Simian T lymphotropic virus) and received a tuberculin test semi-annually.

Infant rhesus macaques are exclusively breastfed for the first 3 months, after which, they are introduced to solid foods and are generally weaned by about 7 months of age. Outdoor-housed rhesus macaques are fed twice daily with Lab Diet, Monkey Diet 5038 (Ralston Purina, St Louis, MO, USA), containing at most 15% crude protein, 5% crude fat, 6% crude fiber, 9% ash, and 12% moisture and after, supplemented daily with fresh fruit and municipal water.

A total of 130 infants were used in this study. Of these 130 infants, 25 were born to dams vaccinated with 40 µg of $H_2O_2$ inactivated *Campylobacter Coli* by intramuscular administration twice during their pregnancy and were themselves subsequently vaccinated with the same vaccine as the dams at 1, 3, and 12 months of age; 28 infants were born to unvaccinated dams and were themselves vaccinated at 1, 3, and 12 months of age; finally,

50 infants were born to unvaccinated dams and were themselves vaccinated at 1, 3, and 5 months of age. An additional 27 infants were born to unvaccinated dams and did not receive any vaccines themselves. Details of the vaccination are included in a separate study[29]. We also utilized fecal samples from the dams of these infants (*n* = 103).

For all infants, fecal swabs were obtained at 1, 3, 6, 9 and 12 months of age as well as upon entry and exit from clinic for diarrhea treatment which consisted of oral hydration, antibiotics, and probiotics administered on a case-by-case basis. Each sample was screened for *Campylobacter coli* and *jejuni* as well as *Shigella flexneri* and *dysenteriae* by microbial culture.

**16S amplicon sequencing.** Total DNA was extracted from rectal swabs using the DNeasy Powersoil Pro Kit (Qiagen, Valencia, CA, USA). The hypervariable V4-V5 region of the 16S rRNA gene was amplified using PCR primers (515F/926R with the forward primers including a 12-bp barcode). PCR reactions were conducted in duplicate and contained 12.5 ml GoTaq master mix, 9.5 ml nuclease-free $H_2O$, 1 ml template DNA, and 1 ml 10 µM primer mix. Thermal cycling parameters were 94 °C for 5 min, 35 cycles of 94 °C for 20 s, 50 °C for 20 s, 72 °C for 30 s, followed by 72 °C for 5 min. PCR products were purified using a MinElute 96 UF PCR Purification Kit (Qiagen, Valencia, CA, USA). Libraries were sequenced (2 ×300 bases) using an Illumina MiSeq.

Raw FASTQ 16S rRNA gene amplicon sequences were uploaded and processed using the QIIME2 analysis pipeline[82]. Briefly, sequences were demultiplexed and the quality filtered using DADA2[83], which filters chimeric sequences and generates an amplicon sequence variant (ASV) table equivalent to an operational taxonomic unit (OTU) table at 100% sequence similarity. Sequence variants were then aligned using MAFFT[84] and a phylogenetic tree was constructed using FastTree2[85]. Taxonomy was assigned to sequence variants using q2-feature-classifier against the SILVA database (release 138)[86]. To prevent sequencing depth bias, samples were rarified to 13,781 sequences per sample before alpha and beta diversity analysis. QIIME 2 was also used to generate the following alpha diversity metrics: richness (as observed ASV), Shannon evenness, and phylogenetic diversity. Beta diversity was estimated in QIIME 2 using weighted and unweighted UniFrac distances[87].

**Shotgun metagenomic library preparation and analysis.** Shotgun metagenomic libraries were prepared from 100 ng of gDNA using the iGenomx RIPTIDE (iGenomx, South San Fransisco CA) per iGenomx recommended protocol and sequenced on an Illumina HiSeq 4000 2 × 100. Raw demultiplexed reads were quality filtered using Trimmomatic[88], and potential host reads were removed by aligning trimmed reads to the *Macaca mulata* genome (Mmul 8.0.1) using BowTie2[89]. Trimmed and decontaminated reads were then annotated using the HUMAnN3 pipeline using default settings with the UniRef90 database and assigned to Metacyc pathways. Functional annotations were normalized using copies per million (CPM) reads before statistical analysis[90–92]. Species-level taxonomy was assigned to quality-controlled short reads using Metaphlan3[32].

**Metagenomic genome assembly and analysis.** Trimmed and decontaminated reads were assembled into contigs using meta-SPAdes with default parameters[93]. Assembled contigs <1 kb were also binned into putative metagenomically assembled genomes (MAGs) using MetaBat[94]. Genome completeness/contamination was tested using CheckM[95], and all bins with a completeness > 80% and contamination < 2% were annotated using PATRIC[96]. The taxonomy of draft genomes was determined using PATRIC's

similar genome finder. For all assembled genomes classified as *Campylobacter* a core gene codon tree was build using PATRICs Phylogenetic tree building tool which used RAxML[97] to align protein sequences of 50 single copy genes found in all genomes included in the analysis.

Pangenomic analysis was conducted using Anvi'o-V7[33]. Assembled fasta files for each genome were first converted to db files using the anvi-script-FASTA-to-contigs-db command, which uses Prodigal[98] to call open reading frames. Each db file was then annotated against the COG database[99] using the anvi-run-ncbi-cogs command. After generating the genome storage file with the anvi-gen-genomes-storage command, the anvi-pan-genome command was used with parameters outlined in the Anvi'o pangenomics tutorial (https://merenlab.org/2016/11/08/pangenomics-v2/). The pangenome was visualized and aesthetics were modified using the *anvi-display-pan* command. GCs found in only a single genome were collapsed into the Singletons bin. GCs found in three or less genomes without a clear pattern, i.e., 3 genomes from 3 different species were collapsed into the Accessory bin to improve the visualization of more abundant GC's. To calculate ANI in Anvi'o, the *anvi-compute-ani* command, which utilizes PyANI[100], was used. To identify functions (i.e., COG annotations) that were differentially distributed among the phylogroups, we used the *anvi-compute-functional-enrichment-in-pan* command as previously described[101]. Taxonomy of these MAGs were further classified using the Genome Taxonomy Database Tool Kit (GTDB-Tk v2.1.1) and GTDB release 214[102,103].

**Statistics and reproducibility**. PERMANOVAs were performed using the R package Vegan[104] function ADONIS. 1-way, non-parametric Kruskal-Wallis ANOVA were implemented using PRISM (V8) to generate *p*-values and utilizing the Dunns post-hoc-test when the initial ANOVA was significant. The linear discriminant analysis effect size (LEfSe) algorithm was used to identify differentially abundant taxa and pathways between groups with a logarithmic Linear discriminant analysis (LDA) score cutoff of 2[105].

**Reporting summary**. Further information on research design is available in the Nature Portfolio Reporting Summary linked to this article.

## Data availability

The datasets supporting the conclusions of this article are available on NCBI's Sequence Read Archive under project numbers PRJNA954012, PRJNA896946, and PRJNA1017507. Supplemental data sets can be found on the online repository Mendeley Data under https://doi.org/10.17632/8w9r67phvt.2 (Supplemental Tables) and https://doi.org/10.17632/7hszwny74w.2 (Supplementary Data 1).

## Code availability

No custom code was developed as a part of this study.

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

## Author contributions

N.S.R., M.K.S., and I.M. conceived and designed the experiments. S.M.H., K.P., A.J.H., and G.E.F. collected samples and provided reagents. N.S.R. and I.R.C. performed the experiments and analyzed the data. N.S.R., I.M. wrote the first draft of the paper. All authors read, edited and approved the manuscript.

## Competing interests

The authors declare no competing interests.
