## [Peer Review File · Communications Biology]

Reviewers' comments:

Reviewer #1 (Remarks to the Author):

In the manuscript by Rhoades et al, the authors monitor microbial compositions of 130 infant rhesus macaques that were part of another publication describing a *Campylobacter coli* vaccination trial (with 27 unvaccinated). Fecal samples were collected at 5 time points during their first year of life. Thirty-four of these animals naturally developed diarrhea and so two additional samples at clinic entry and exit were collected together with the other samples obtained during the study. The authors discovered that infants that subsequently developed diarrhea had higher abundances of *Campylobacter* prior to demonstrating clinical signs of illness. As they stated, "these findings suggest that subclinical colonization with pathobionts may disrupt the maturation of the microbiome and precede clinical disease." Also, the authors noted that the microbiome of dams, who gave birth to macaques that later experienced diarrhea, showed increased *E. coli*, *Bifidobacterium* and *Campylobacter* when shotgun metagenomes were compared.

They then constructed 36 *Campylobacter* metagenomic assembled genomes (MAGs), and 15 of these MAGs belonged to 4 groups of uncultured *Campylobacter* species. The authors provide an extensive gene content comparison with known species which will need to be followed up in future studies experimentally, since as they admit, these microbes are unculturable, so they were unable to confirm these differences phenotypically or whether these differences were due to low homology or lower quality/incomplete metagenome sequences.

Comments

The authors mention on Page 7, lines 160-161 that "C. coli vaccination reduced overall diarrheal disease burden, it did not impact the composition of the fecal microbiome". Additional details about the vaccination trial would be useful to better understand the results. Were the infants also vaccinated with the inactivated vaccine intramuscularly or orally? Was the C. coli vaccine strain marked so that the authors could determine that no escape survivors were being detected in the MAGs where C. coli was detected?

The increased folate transformation link with healthy infants is interesting (page 10, lines 210-214). Did the authors try to test reference strains of *Bracyspira* or *Prevotella* to demonstrate this link – or find other reports that support this association with these microbes (besides the histopathological findings from infant macaques with EED)?

Similarly, on page 10, lines 228-230, the microbiomes of infants that remained asymptomatic showed higher abundance of *Lactobacillus*, *Treponema*, *Bracyspira* species, and *Helicobacter macacae*. Looking at the authors' previous publications, *H. macacae* was shown before to be in higher abundance in non-human primates without diarrhea – and this species is unique to macaques. Is the relationship with the other genera present in healthy human infants compared to those with diarrhea?

Also, why do the authors think *C. coli* is more prevalent compared to *C. jejuni* in rhesus infants (based on first suppl table), while the opposite is observed in human infants?

Page 10/11, lines 233-235, the authors mention that the abundance of *Prevotella copri*, *Streptococcus lutetuensis*, *Veillonella parvula*, and *Campylobacter* is increased after exit from the clinic. Could the authors speculate in the Discussion what they think is happening with *Campylobacter* – do they believe the pathobiont levels are continuing to decrease, or do they suspect that the infants will become asymptomatic carriers after disease symptoms resolve and then the cycle will be repeated with dams having higher *Campylobacter* levels that subsequently have infants with higher probabilities for developing diarrhea?

Page 11, line 246 (and in methods): the phylogenetic tree was based on the alignment of 50

conserved single copy
genes – the identity of these 50 genes should be listed in supplementary.

Page 13, line 285: the authors state that *Campylobacter* are motile due to a single polar flagellum – this should be motile due to a single flagellum at both poles.

The authors should check the spelling of *Candidatus* and of *C. infans* throughout the manuscript.

The catalase gene is *katA* not *katE*.

On page 15, lines 325-326, the authors state that the human reference genomes of both *C. coli* and *C. hyointestinalis* lack *cdtA* and *cdtC*. Is the 'human reference' *hyointestinalis* genome LMG 9260? From that annotation, all 3 *cdt* genes are present. Also, 3 of the 4 *C. hominis* genomes have all 3 genes. One *lawsonii* has *cdtC*, but carries mutations in *cdtA* and *cdtB*. I believe all *C. coli* strains are *cdt+*. The only *C. jejuni* strains that tend to be consistently *cdt-* are the subspecies *doylei*. Are all the strains used for comparison those that are in the trees in Fig S5 (this should be emphasized, if so)? The authors should double-check their comparisons.

Some of the figures need to be improved by increasing font sizes and providing a noticeably different color for bacteria that are being highlighted in the results/discussion and/or the authors should provide tables listing bacteria and their percent abundance in the supplementary. For example, I could not distinguish the difference between *Campylobacter* and *H. mucacae* in Figure 1 even when the figure is full screen. Similarly, the different shades of blue cannot be distinguished in Figure 2. Also, UMG-1 versus UMG-2 are difficult to distinguish in Figure 3, and Figure 4 is too small to make any sense out of it – the analyses would be much more informative if the data was put into a table listing all the reference genomes, genes, etc. In Figure 3, and in other legends, DX should be defined each time so the reader does not have to search in the text. In Figure 5D, what is happening at the bottom? Has the complex I horizontal title covered up words that should appear vertically?

I did not see a link for the MAG data, the metagenomes should be deposited at:
<https://www.ncbi.nlm.nih.gov/genbank/metagenome/>

The authors should also check that the spelling of *C. jejuni* and *C. coli* in italics, rather than *C. Jejuni* and *C. Coli* or *c. jejuni* and *c. coli* in the text and figures.

Reviewer #2 (Remarks to the Author):

Q.What are the major claims of the paper?

Response: The manuscript has explored the topic of the association between poor development of young children's gut microbiome that are inflamed with *Campylobacter*. The subject has gained worldwide Epidemiology/Public Health related scientific community's attention, especially during the last 5 years, after the publication of MALED results. The study adds some very meaningful and key insights into the affect of campylobacter and diarrheal incidences on the temporal progression of a normal gut microbiome. The use of marcus monkey model is a convenient and intelligent choice as the implementation of such a longitudinal study with human infants is very complex both in terms of financial and cultural aspects. However, the findings highlighted in this work repostulates the earlier literature as well point to the specific mechanism leading to the cause of underdevelopment.

Overall interesting key findings:

- Using 16 S metagenomic sequencing authors have shown in detail, that, how diseased infants drifted apart from the normal trajectory of gut microbiome development and resulted in stunting and delayed maturation, increased dissimilarity with age and less phylogenetic diversity than a non-sick cohort.

Detailed analysis also showed the colonization of pathogenic bacterial Genera, including *Prevotella*, *Streptococcus*, and *Campylobacter* species, in sick infants' gut with age after the diarrheal event. Microbiome stunting, reduced community richness, and increased *Campylobacter* abundance were three major highlights of 16S analysis.

- Authors also supported their 16S finding with mycobiome profiling where they observed similar patterns of reduced diversity in sick infants over 12 months.
- Authors also investigated the functional maturation of the gut microbiome using advanced shotgun sequencing metagenome data and found the enrichment of anhydro-fructose and amino-butyrate degradation pathways in diarrheal infants. Also, clear-cut difference in the functional pathways was observed in sick and non-sick infants.
- Detailed taxonomic maturation differences were also analyzed via shotgun data. Among many contrasting genus presences, notably, authors observed the increased relative abundance of *Anaerostipes hadrus* in diarrhea-developing infants, along with an association of multiple species of *Campylobacter* and *Prevotella* with acute diarrheal cases.
- Authors also assembled metagenomic genomes (MAGs) for *Campylobacter* species related to diarrhea and also reported the presence of some newly identified uncultured species that were mapped closely to recently reported diarrheal *Campylobacter* species including *C. infans*, *C. hyointestinalis* and *C. troglodytis*, along with well-known *C. coli/jejuni*.
- Authors compared the Genomic diversity of unknown MAGs with genomes from macaques and human MAGs along with multiple *Campylobacter* references to establish global relatedness, validity, and usefulness of the animal model for this study.
- In addition, the authors performed a detailed analysis to identify virulent factors inclusion adhesion and motility in the newly identified *Campylobacter*. The findings increase the weight and importance of the study in providing in-depth insights of the pathogenetic mechanisms of newly found *Campylobacter* species in this study.

Q. Are they novel and will they be of interest to others in the community and the wider field?

Response: The claims/findings presented in this paper are critical, especially for the researcher trying to relate the effect of different enteric pathogens on young children's gut microbiome development and its impact on their growth especially in LMIC where the child stunting rate is very high. Such work will also help studies trying to pave the way for tracing the proper routes of infection and intervention strategies to alleviate young population stunting in the longer term.

Q. If the conclusions are not original, it would be helpful if you could provide relevant references.

Response: Conclusions are based on original research and well cross-referenced to the latest research. Elaborated cross-referencing has been included and relevant comparison between monkey and human infants has been shown and minor limitations are well documented.

Q. Is the work convincing, and if not, what further evidence would be required to strengthen the conclusions?

Response: The work looks quite convincing. Design and Implementation are very logical and well-connected. Analysis methods and pipelines are also appropriately detailed.

Q. On a more subjective note, do you feel that the paper will influence thinking in the field?

Response: This work will be a great addition to the existing literature on *Campylobacter*'s role in poor gut microbiome development and modulating child growth.

We thank the reviewers for their thoughtful and thorough review of the manuscript. We provide a response to their concerns below and have made edits to the text that are highlighted in yellow for ease of readability.

Reviewer #1

The authors mention on Page 7, lines 160-161 that “*C. coli* vaccination reduced overall diarrheal disease burden, it did not impact the composition of the fecal microbiome”. Additional details about the vaccination trial would be useful to better understand the results. Were the infants also vaccinated with the inactivated vaccine intramuscularly or orally? Was the *C. coli* vaccine strain marked so that the authors could determine that no escape survivors were being detected in the MAGs where *C. coli* was detected?

We apologize for the oversight. Since the submission of this manuscript, the paper describing the vaccine formulation and outcomes has been accepted. We now include a reference to that paper in the methods section that would allow the reader to obtain all the information surrounding the vaccination. Briefly, infants were vaccinated with an inactivated *E. Coli* isolated from a sick macaque intramuscularly. The MAGs do not represent vaccine strain that escaped inactivation.

The increased folate transformation link with healthy infants is interesting (page 10, lines 210-214). Did the authors try to test reference strains of *Bracyspira* or *Prevotella* to demonstrate this link – or find other reports that support this association with these microbes (besides the histopathological findings from infant macaques with EED)?

Unfortunately, little information has been reported on the implications of *Bracyspira* on folate transformation. However, one published article reported that several species of *Bifidobacterium* (including *B. pseudocatenulatum*, *B. adolescentis*, and *B. longum*) are involved in folate synthesis. These points have been added to the discussion.

Similarly, on page 10, lines 228-230, the microbiomes of infants that remained asymptomatic showed higher abundance of *Lactobacillus*, *Treponema*, *Bracyspira* species, and *Helicobacter macacae*. Looking at the authors’ previous publications, *H. macacae* was shown before to be in higher abundance in non-human primates without diarrhea – and this species is unique to macaques. Is the relationship with the other genera present in healthy human infants compared to those with diarrhea?

Lactobacillus is a prominent genera in a healthy human infant gut, specifically in those born vaginally. Conversely, that is reduced in infants with diarrhea. These points have been added to the discussion.

Also, why do the authors think *C. coli* is more prevalent compared to *C. jejuni* in rhesus infants (based on first suppl table), while the opposite is observed in human infants?

This is a very important observation brought up by the reviewer. Within humans, the CDC reports 90% of *Campylobacter* are *C. jejuni* while the rest are *C. coli* and other species. However, two papers published in Peru and Egypt reported 30-37% *C. coli* as the causative species in *Campylobacter*-associated disease. Colleagues at the Oregon National Primate Research Center and the California Primate Research Center report different proportions of *C. coli* and *C. jejuni* circulating among their primates. We speculate that a “founder” effect may influence *Campylobacter* dominance in which the first species that establishes itself within a population or community becomes the majority.

Page 10/11, lines 233-235, the authors mention that the abundance of *Prevotella copri*, *Streptococcus lutetensis*, *Veillonella parvula*, and *Campylobacter* is increased after exit from the clinic. Could the authors speculate in the Discussion what they think is happening with *Campylobacter* – do they believe

the pathobiont levels are continuing to decrease, or do they suspect that the infants will become asymptomatic carriers after disease symptoms resolve and then the cycle will be repeated with dams having higher *Campylobacter* levels that subsequently have infants with higher probabilities for developing diarrhea?

Campylobacter-associated infections are generally self-limiting, so levels of these species are likely to decrease following acute disease. We don't have the ability to determine if *Campylobacter* is completely cleared upon recovery, but it is more likely that individuals become asymptomatic carriers. This may contribute to the re-occurrence or spread of the disease. This is now addressed in the discussion.

Page 11, line 246 (and in methods): the phylogenetic tree was based on the alignment of 50 conserved single copy

genes – the identity of these 50 genes should be listed in supplementary.

This has been included as a new supplemental table (Table S7).

Page 13, line 285: the authors state that *Campylobacter* are motile due to a single polar flagellum – this should be motile due to a single flagellum at both poles.

This correction has been addressed in the text.

The authors should check the spelling of *Candidatus* and of *C. infans* throughout the manuscript.

We apologize for the inconsistency in the spelling- this has now been addressed.

The catalase gene is *katA* not *katE*.

We thank the reviewer for bringing this to our attention. This correction has been made throughout the figures and manuscript.

On page 15, lines 325-326, the authors state that the human reference genomes of both *C. coli* and *C. hyointestinalis* lack *cdtA* and *cdtC*. Is the 'human reference' *hyointestinalis* genome LMG 9260? From that annotation, all 3 *cdt* genes are present. Also, 3 of the 4 *C. hominis* genomes have all 3 genes. One *lawsonii* has *cdtC*, but carries mutations in *cdtA* and *cdtB*. I believe all *C. coli* strains are *cdt+*. The only *C. jejuni* strains that tend to be consistently *cdt-* are the subspecies *doylei*. Are all the strains used for comparison those that are in the trees in Fig S5 (this should be emphasized, if so)? The authors should double-check their comparisons.

We have qualified the statement of "human reference genomes" to now be "human representatives.". The reference genome for the species "*Campylobacter hyointestinalis* subsp. *lawsonii* strain CHY5" was isolated from a pig and we thought it was more relevant to include a human pathogen "*C.*

hyointestinalis subsp. *hyointestinalis* LMG 9260" The *Campylobacter* strains used for this comparison can now be found in Table S8.

Some of the figures need to be improved by increasing font sizes and providing a noticeably different color for bacteria that are being highlighted in the results/discussion and/or the authors should provide tables listing bacteria and their percent abundance in the supplementary. For example, I could not distinguish the difference between *Campylobacter* and *H. mucacae* in Figure 1 even when the figure is full screen. Similarly, the different shades of blue cannot be distinguished in Figure 2. Also, UMG-1 versus UMG-2 are difficult to distinguish in Figure 3, and Figure 4 is too small to make any sense out of it – the analyses would be much more informative if the data was put into a table listing all the reference genomes, genes, etc. In Figure 3, and in other legends, DX should be defined each time so the reader does not have to search in the text. In Figure 5D, what is happening at the bottom? Has the complex I horizontal title covered up words that should appear vertically?

Font sizes have been increased in figures, so that they are more legible. Colors in Figures, especially in taxa plots have been altered to be more distinguishable. DX is now defined in all legends. In Figure 5D the horizontal title is not covering anything. New supplemental table of relative abundance of top 30 bacteria in Fig 1C has been added (Table S2).

I did not see a link for the MAG data, the metagenomes should be deposited at:

The MAG data has been deposited into the SRA under project number PRJNA1017507. This information has been included in the manuscript.

The authors should also check that the spelling of *C. jejuni* and *C. coli* in italics, rather than *C. Jejuni* and *C. Coli* or *c. jejuni* and *c. coli* in the text and figures.

The spelling of these species is now consistent.

Reviewers' comments:

Reviewer #1 (Remarks to the Author):

The reviewers have addressed all the points that I raised and so I have no further concerns.

We thank the editor for their thoughtful and thorough review of the manuscript. We provide a response to their concerns below and have made edits to the text that are highlighted in yellow for ease of readability.

1. While the reviewer did not have any further requests, it came to our attention that the taxonomy assignment seems to be outdated, since the phylum "Epsilonproteobacteria" is not valid anymore (see <https://www.microbiologyresearch.org/content/journal/ijsem/10.1099/ijsem.0.005056>). Therefore, we strongly suggest to use an updated taxonomic classifier e.g. GTDB-tk, see <https://academic.oup.com/bioinformatics/article/38/23/5315/6758240> and <https://academic.oup.com/bioinformatics/article/36/6/1925/5626182> or metaphlan 4 if you want to stick with metaphlan, <https://www.nature.com/articles/s41587-023-01688-w> - I highly recommend GTDBtk with the current release R214 though.
 - a. We thank you for bringing this to our attention. Phyla names in Figure 1C and throughout the manuscript have now been updated to the current accepted nomenclature.
 - b. MAG taxonomy was also classified using GTDB-Tk v2.1.1 using the Genome Taxonomy Database R214. Table S6 now includes the results. References for GTDB-Tk and its database have also been included in the references.
2. Also please double check references - Ref 32 is not Metaphlan (Line 256). Ref 94 describes MetaPhlan 1, not version 3.
 - a. We thank you for catching this error. Ref 32 has now been replaced to cover Metaphlan 3. Ref 94 was also removed.